# A Competitive Sprinter’s Resting Blood Lactate Levels Fluctuate with a One-Year Training Cycle: Case Reports

**DOI:** 10.3390/jfmk6040095

**Published:** 2021-11-22

**Authors:** Ryotaro Kano, Kohei Sato

**Affiliations:** 1Graduate School of Education, Tokyo Gakugei University, Tokyo 184-8501, Japan; ryotaro172607@gmail.com; 2Department of Health and Physical Education, Tokyo Gakugei University, Tokyo 184-8501, Japan

**Keywords:** training periodization, athlete, metabolic characteristics

## Abstract

It has been reported that the variability of resting blood lactate concentration (BLa) is related to metabolic capacity. However, it is unclear whether the resting BLa of athletes can be utilized as a metabolic biomarker. This longitudinal case study tested the hypothesis that resting BLa levels in the morning fluctuate with a 1-year training cycle. The subject was an adult male sprinter, and BLa and blood glucose at the time of waking were measured every day for 1 year. The training cycles were divided into five phases: 1. Basic training: high-intensity and high-volume load; 2. Condition and speed training: high-intensity and low-volume load; 3. Competition training I: track race and high-intensity load; 4. Conditioning for injury; 5. Competition training II. The mean BLa levels in the basic training (1.10 ± 0.32 mmol/L and competition training I (1.06 ± 0.28 mmol/L) phases were significantly lower than in the condition and speed training (1.26 ± 0.40 mmol/L) and conditioning injury (1.37 ± 0.34 mmol/L) phases. The clarified training cycle dependence of resting BLa is suggested to be related to the ability to utilize lactate as an energy substrate with fluctuations in oxidative metabolic capacity. This case report supports the tentative hypothesis that resting BLa may be a biomarker index linked to the metabolic capacity according to the training cycle.

## 1. Introduction

It is known that the specific variability of blood lactate concentration (BLa) at rest can be utilized as a metabolic biomarker [1]. Previous studies have elucidated specific high BLa at rest in metabolic disorders such as diabetes [2,3], obesity [4], and chronic fatigue syndrome [5,6]. Pathological mitochondrial dysfunction is a sequela of diabetes and induces a decrease in the oxidative ATP production capacity [3]. In parallel with this metabolic state, glycolysis-dependent ATP production increases lactic acid production as a compensatory effect. Suppression of oxidative energy production capacity is reflected as a continuous increase in blood lactate level at rest [2,3]. In addition, the high level of BLa at rest observed in patients with severe sepsis and heart disease is a diagnostic criterion reflecting an adverse prognosis [7,8]. In contrast to clinical application as a diagnostic criterion for metabolic diseases, resting BLa in athletes has not received focused attention as a biomarker. Although there are many research findings that elucidate exercise intensity-dependent metabolic changes, such as lactate threshold during exercise, few studies have evaluated the BLa variability at rest in athletes. Exercise training and conditioning performed by athletes alters mitochondrial energy metabolism characteristics [9]. In addition, training increases the amount of mitochondria and MCTs (monocarboxylate transporters), which may have a significant effect on blood lactate levels. Therefore, it is possible that the lactate metabolism characteristics related to the training state are reflected in the variability of resting BLa. 

Recently, a study evaluating the physiological conditioning of professional cycle athletes based on morning heart rate and blood pressure variability was published [10]. The present study may contribute to the evaluation of physiological conditioning by resting morning BLa monitoring. To verify whether resting BLa reflects training-dependent changes, individual longitudinal observations based on long training cycles are required to avoid the effects of individual acute exercises. Therefore, we tested the hypothesis that resting morning BLa levels fluctuate systematically within a 1-year training cycle by conducting a longitudinal case study of an adult 200 m sprinter. The subject, a 200 m track sprinter, was selected as a representative subject to perform mixed aerobic and anaerobic exercises dependent on the long-term training cycle.

## 2. Materials and Methods

Subject: The subject was a healthy adult male sprinter (22 years old; 200 m 21.48 s; 400 m 48.54 s). As regards physical properties (height 178.0 cm; weight 67.0 kg), the weight variation of the experimental period was within 6.9%. There were no smoking and drinking habits or medication. The study was approved by the Tokyo Gakugei University Ethics Board. (NO.506). 

Training period setting: The 1-year training period was set to 5 periods, which is the sum of 4 planned cycles and a training discontinuation period due to injury as follows. 1. Basic training (2019/November~2020/February); middle- or high-intensity and high-volume training (tempo running, interval running, circuit training, long jogging etc.). 2. Condition and speed training (2020/March~April); high-intensity and low-volume training (short sprint time trial, start etc.). 3. Competition training I (2020/April~July; record for this period was 200 m 21.63 s); track race, high-intensity middle-volume training (all out sprint training, start dash, etc.). 4. Conditioning for injury (2020/August); no running. 5. Competition training II (2020/September~2020/October; record for this period was 200 m 22.16 sec); track race after failed recovery. Sprint running (start: 0–30 m, short: 30–150 m, and long sprint: 150–500 m) distance, except for the warm-up and cooling-down running, was recorded. The outside temperature (mean, maximum, and minimum; °C) for each month at Tokyo were as follows: October 2019 (19.4, 23.3, 16.4), November (13.1, 17.7, 9.3), December (8.5, 12.6, 5.2), 2020 January (7.1, 11.1, 3.7), February (8.3, 13.3, 4.0), March (10.7, 16.0, 6.2), April (12.8, 18.2, 7.9), May (19.5, 24.0, 15.6), June (23.2, 27.5, 19.8), July (24.3, 27.7, 21.8), August (29.1, 34.1, 25.3), September (24.2, 28.1, 21.5), October (17.5, 21.4, 14.4), November (14.0, 18.6, 10.1), and December (7.7, 12.3, 3.7).

Measurement: The period of this case study was from November 2019 to October 2020. Immediately after waking in the morning, blood samples were taken by puncturing the tip of the left middle finger (Naturalet EZ, Arkray, Tokyo, Japan). Blood lactate and glucose were determined by a lactate analyzer (Lactate Pro2, Arkray) and a glucose analyzer (PG-7320, Arkray), respectively. 

Statistical analyses: Values are expressed as means ± SD. Statistical analyses were performed in Prism version 7.0 (GraphPad Software, San Diego, CA, USA). The changes in BLa and glucose level were compared within each training period using a one-way ANOVA with followed by Tukey’s post hoc test. The level of significance was set at *p* < 0.05.

## 3. Results

Figure 1 shows changes in sprint running distance, BLa and glucose levels every morning for 1-year. The average value of the total period of sprint running distance (347 ± 578 m/ day), BLa (1.17 ± 0.34 mmol/L), and glucose (90.3 ± 6.1 mg/dL is shown by the red dashed line. Figure 2 shows the average value for each training period. For BLa according to the training cycle, basic training (1.10 ± 0.32 mmol/L) and competition training I (1.06 ± 0.28 mmol/L) were significantly lower than conditioning and speed training (1.26 ± 0.40 mmol/L) and conditioning for injury (1.37 ± 0.34 mmol/L). The waking blood glucose level of basic training (94.4 ± 8.7 mg/dL) was significantly higher than that of competition training I (88.5 ± 5.0 mg/dL), conditioning injury (87.5 ± 4.1 mg/dL), and competition training II (89.0 ± 6.1 mg/dL). 

## 4. Discussion

This case study, which involved a longitudinal follow-up of the athlete, revealed that the resting BLa level fluctuates with the training cycle. Interestingly, during the qualitative and quantitative high-load training cycle, the resting BLa levels were significantly reduced, and conversely, continuously high BLa levels were observed during the reduced-load training period (i.e., conditioning and speed training and conditioning for injury). It is possible that these changes caused by the training cycle reflect the metabolic conditioning of the athlete. Mitochondrial volume and function indicate a short-term response to the training cycle. For example, 6 weeks of endurance training increased mitochondrial volume by about 80%, while 3 weeks of detraining decreased that volume by about 34% [11]. These observations suggest that metabolic conditioning associated with mitochondrial function depends on a training cycle of several weeks. Basic training and competition training were mainly high-intensity exercise, such as high-intensity interval exercise and sprint interval running. Interval training is known to effectively enhance mitochondrial function [12,13]. The functional enhancement of mitochondria by training increases the efficiency of using lactic acid as an energy substrate [9]. Moreover, several intense interval training bouts increase the content of both MCT1 and MCT4 in human skeletal muscle [14]. It has been reported that MCT1, which plays a role in lactate uptake into muscle and mitochondria, has a greater adaptability than MCT4 after high-intensity exercise training [15]. Therefore, the low BLa level at the time of waking observed during the basic and competition training periods may, potentially, be related to the training cycle. On the other hand, high BLa levels continued during the conditioning period and ligament injury period. The resting BLa variability shown by this longitudinal observation may be a biomarker for monitoring training cycle-dependent metabolic function.

The cycle change in blood glucose concentration was independent of the change in BLa level. Epidemiological studies have reported that fasting blood glucose levels tend to be higher in winter [16,17]. The fasting blood glucose level of this subject was also significantly higher during the winter training phase.

This case study has some limitations. First, it remains unclear whether the transient effects of training are reflected in BLa levels the following morning. Although the BLa recovery process after high-intensity exercise is relatively fast (recovery time to 2.5 mmol value time ~21–30 min) [18], the influence on lactate metabolism at rest is not clear. Extremely high-intensity training may induce high levels of BLa the following morning. Increased sympathetic nervous system activity during athletic competition may affect the secretion and blood concentration of glycolysis-stimulating hormones such as adrenaline. The sudden appearance of higher BLa levels observed in the basic training and competition training periods may be influenced by these sympathetic factors and mental- and fatigue-related factors. Second, the relationship between training cycle-dependent fluctuations in resting lactate metabolism and exercise performance was not explained in this study. Elucidation of the relationship between endurance performance and resting BLa levels contributes to athletes’ training strategies and conditioning. Third, the presence of individual differences for training adaptation were not studied. For example, in the 6-week sprint interval training period, the increase rate of MCT1 protein expression was in the range of 30% to 530% [13]. Future studies should focus on elucidating the physiological variability with genetic features such as muscle fiber composition.

In addition, it is important to elucidate the specificities of sporting events, such as the differences between sprint and endurance athletes. These are future research perspectives for practical application to various athletes.

## 5. Conclusions

The findings in this case report support the existence of training cycle-dependent fluctuations in resting lactate metabolism. It is proposed that such fluctuations in resting lactic acid concentrations may have the potential to serve as a convenient and insightful athletic biomarker.

## Figures and Tables

**Figure 1 jfmk-06-00095-f001:**
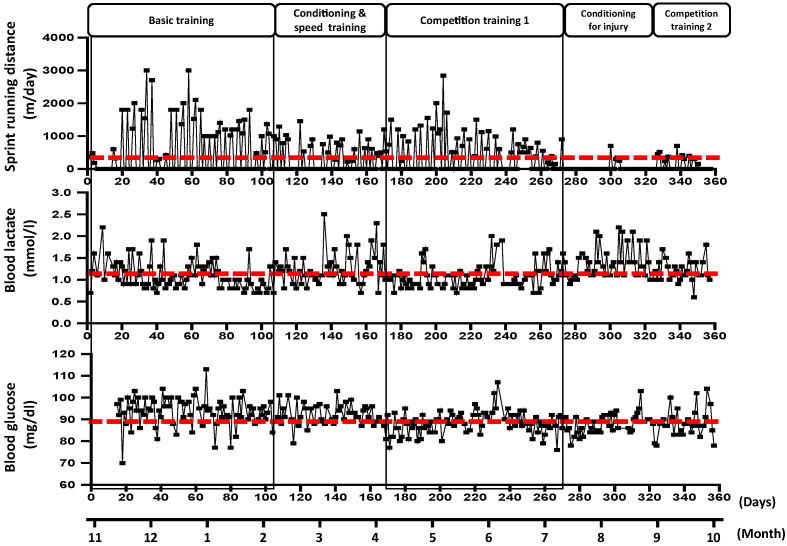
Sprint running distance (**upper graph**) and changes in daily morning blood lactate (**middle graph**) and glucose (**lower graph**) concentrations. The average value for all training period is shown by the red line.

**Figure 2 jfmk-06-00095-f002:**
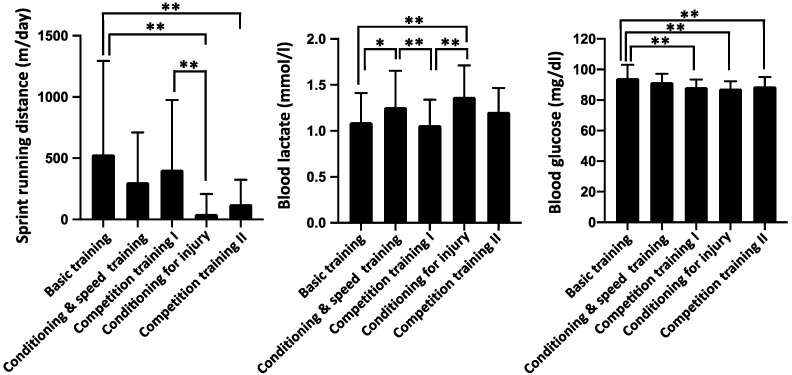
Comparison between sprint running distance, blood lactate and glucose levels in each training period. values are presented as means with SD. One-way ANOVA with Tukey’s post-hoc analyses for each training period and *p*-values were summarized as follows: * *p* < 0.05, ** *p* < 0.01.

## Data Availability

All data generated or analyzed during this study are included in this article.

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
