# Peer review of "A Competitive Sprinter’s Resting Blood Lactate Levels Fluctuate with a One-Year Training Cycle: Case Reports"

_jfmk, 2021, doi:10.3390/jfmk6040095_

Round 1
Reviewer 1 Report
This is a very interesting case report about blood lactate levels as a possibile biomarker for athlete's status. Topic is interesting for practical application and english is fluent for the reader.
I suggest you to improve you article following CARE guidelines checklist for case report (https://www.care-statement.org/checklist). In this way, you'll check if everything is ok and if there are some sections that can be improved.
Pay attention to line 158: it should be deleted!
Moreover, I suggest you the following reference about the monitoring of recover status in athletes, since it can highlight how this is an hot topic in literature! --> https://www.frontiersin.org/articles/10.3389/fspor.2020.00067/full
Author Response
This is a very interesting case report about blood lactate levels as a possibile biomarker for athlete's status. Topic is interesting for practical application and english is fluent for the reader.
We would like to take this opportunity to extend our sincere thanks to the editor and reviewers for taking the time to scrutinize our manuscript. We agree with your suggestion, and thus we have provided a brief response to each of the points below including modifications to the revised text. We believe that the manuscript has substantially been improved with the comments and suggestions.
I suggest you to improve you article following CARE guidelines checklist for case report (https://www.care-statement.org/checklist). In this way, you'll check if everything is ok and if there are some sections that can be improved.
Thank you so much for your important comment.Referring to the guidelines for this case report, I made some modifications to the title and the selection of subject (P1,L 2~3).
Pay attention to line 158: it should be deleted! 
Thank you for pointing out the careless mistakes. The sentence has been deleted.
Moreover, I suggest you the following reference about the monitoring of recover status in athletes, since it can highlight how this is an hot topic in literature! --> https://www.frontiersin.org/articles/10.3389/fspor.2020.00067/full
Thank you for the information on this interesting case report literature. We have cited this reference and added it to the introduction (P 2,L 46~49).
“ Recently, an attempt to evaluate the physiological conditioning of professional cycle athletes based on morning heart rate and blood pressure variability has been reported [10]. the present study may contribute to the evaluation of physiological conditioning by resting morning BLa monitoring.”
Reviewer 2 Report
The manuscript is of potential interest, but some major changes are needed before considering it suitable for publication.
The introduction should explain better how and why the method proposed by the authors could be an opportunity in field practice with athletes. In other words, why the lactate concentration should fluctuate, and how and why such a fluctuation could be relevant?
Another point is the choice of the subject. Why this subject? Why this sport specialty? What if an endurance athlete is tested?
The discussion should therefore be revised to respond to the aforementioned questions.
Author Response
The manuscript is of potential interest, but some major changes are needed before considering it suitable for publication.
Thank you so much for your careful reviewing our manuscript and constructive comments. We have revised this manuscript as you suggested. We hope that this revision is improved and that you are satisfied with the changes made.
The introduction should explain better how and why the method proposed by the authors could be an opportunity in field practice with athletes. In other words, why the lactate concentration should fluctuate, and how and why such a fluctuation could be relevant?
Thank you so much for your important comment. We have added more information for this reviewer’s concern. Training increases the amount of mitochondria and MCTs (monocarboxylate transporters). They may have a significant effect on blood lactate levels. (P1, L 41~43).
Another point is the choice of the subject. Why this subject? Why this sport specialty? What if an endurance athlete is tested?
We understand your concern. In general, endurance athletes have a lower percentage of anaerobic training and less variability in their training cycles. With this background, we focused on subject of sprint events. The subject, a 200 m track sprinter, was selected as a representative subject to perform a mixed aerobic and anaerobic exercise dependent on the long-term training cycle. (P 2 L 53~55) (P4 L 162~164)
The discussion should therefore be revised to respond to the aforementioned questions.
Thank you so much for your important comments. The authors believe that specific recommendations on how athletes can use resting lactate variability as a practical biomarker are important. On the other hand, recommendations on how to use lactate as a biomarker are considered to be a subject for future research. Therefore, we have added the following comments to the discussion(P4,L 162~164). “In addition, it is important to elucidate the specificities of sporting events, such as the differences between sprint and endurance athletes. These are future research perspectives for practical application to various athletes.”
We believe that these additions/revisions have increased the scope, mechanistic insight and clarity of the manuscript along with its potential impact.
Round 2
Reviewer 2 Report
I have no further comments